# Validation of Artificial Intelligence (AI)-Assisted Flow Cytometry Analysis for Immunological Disorders

**DOI:** 10.3390/diagnostics14040420

**Published:** 2024-02-14

**Authors:** Zhengchun Lu, Mayu Morita, Tyler S. Yeager, Yunpeng Lyu, Sophia Y. Wang, Zhigang Wang, Guang Fan

**Affiliations:** 1Department of Pathology and Laboratory Medicine, Oregon Health & Science University, 3181 SW Sam Jackson Park Road, Portland, OR 97239, USA; luzh@ohsu.edu (Z.L.); morita@ohsu.edu (M.M.); yeagert@ohsu.edu (T.S.Y.); lyp141125@gmail.com (Y.L.); sophiawangyn@gmail.com (S.Y.W.); 2DeepCyto LLC, West Linn, OR 97068, USA; alan@deep-cyto.com

**Keywords:** artificial intelligence, multiparameter flow cytometry, clinical cases, panel for autoimmune lymphoproliferative syndrome, hematopathology

## Abstract

Flow cytometry is a vital diagnostic tool for hematologic and immunologic disorders, but manual analysis is prone to variation and time-consuming. Over the last decade, artificial intelligence (AI) has advanced significantly. In this study, we developed and validated an AI-assisted flow cytometry workflow using 379 clinical cases from 2021, employing a 3-tube, 10-color flow panel with 21 antibodies for primary immunodeficiency diseases and related immunological disorders. The AI software (DeepFlow™, version 2.1.1) is fully automated, reducing analysis time to under 5 min per case. It interacts with hematopatholoists for manual gating adjustments when necessary. Using proprietary multidimensional density–phenotype coupling algorithm, the AI model accurately classifies and enumerates T, B, and NK cells, along with important immune cell subsets, including CD4+ helper T cells, CD8+ cytotoxic T cells, CD3+/CD4−/CD8− double-negative T cells, and class-switched or non-switched B cells. Compared to manual analysis with hematopathologist-determined lymphocyte subset percentages as the gold standard, the AI model exhibited a strong correlation (*r* > 0.9) across lymphocyte subsets. This study highlights the accuracy and efficiency of AI-assisted flow cytometry in diagnosing immunological disorders in a clinical setting, providing a transformative approach within a concise timeframe.

## 1. Introduction

Multiparameter flow cytometry (MFC) stands as one of the most commonly employed diagnostic tools in hematopathology. This high-dimensional technique offers sensitive detection and quantification of antigen expression on individual cells through fluorescent antibody staining [1]. Primary immunodeficiency diseases (PIDs) constitute a group of inherited genetic disorders affecting the immune system, including X-linked agammaglobulinemia (XLA), severe combined immunodeficiency (SCID), autoimmune lymphoproliferative syndrome (ALPS), DiGeorge syndrome, common variable immunodeficiency (CVID), and other immunologically related deficiencies. Due to their broad and sometimes overlapping presentations, the clinical suspicion of PID typically arises from a thorough review of the patient’s clinical history, focusing on recurrent infections, autoimmunity, and other complications. Although a definitive diagnosis relies on genetic testing, which can be time-consuming and expensive, major forms of PIDs can exhibit unique immune cell deficiencies quickly identified by flow cytometry [2]. For instance, XLA, characterized by the absence of circulating B cells and a severe reduction in serum immunoglobulins due to the *BTK* gene mutation can be detected by the absence of B cells in peripheral blood. Furthermore, XLA carriers can also be identified by flow cytometry using monocytes [3]. Autoimmune lymphoproliferative syndrome (ALPS) is a rare genetic disorder of lymphocyte apoptosis characterized by early-onset lymphadenopathy, hepatosplenomegaly, autoimmune cytopenia, and an increased risk of lymphoma [4]. One of the diagnostic hallmarks of ALPS is the accumulation of CD4/CD8 double-negative T cells (DNTs) that express T-cell receptor αβ. The quantification of TCRαβ+/DNT > 1.5% of total lymphocytes or >2.5% of CD3+ lymphocytes is part of the essential diagnostic criteria of ALPS [5]. The DNT population can be easily recognized by flow cytometry. DiGeorge syndrome is a congenital immune deficiency disorder, mainly due to abnormal T-cell maturation and dysfunction, with a spectrum of abnormally low or high levels of lymphoid subsets [6]. The characteristic feature is significantly decreased or absent CD45RA+ T cells and decreased switched memory B cells, causing decreased immune function and presenting infectious and decreased immunoglobin levels. While molecular and genetic analysis is the ultimate approach to establishing the diagnosis of PIDs, flow cytometry has become a critical component for diagnosing and evaluating the immunologic function of these patients.

In our practice, we use a 10-color flow cytometry analysis panel comprising 26 antibodies to evaluate PIDs for pediatric and adult patients [7,8,9]. This panel is also used to assess lymphocyte reconstitution and lymphocyte activation after bone marrow or stem cell transplantation [10]. Whole blood is tested in three flow cytometry tubes with a panel of T-, B-, and NK-cell markers, including CD3, CD4, CD8, CD13, CD16, CD25, CD127, TCRαβ, TCRγδ, CD56, IgD, and others. The raw data for each sample are collected in the linear mode data (LMD) file format. The analysis of these data requires multiple manual steps by technologists, namely creating a case folder; transferring data to a designated server; adjusting the gating of lymphocyte subsets via Kaluza^TM^ software (version 2.2.1) by Beckman Coulter, Inc. (Brea, CA, USA); and saving the Kaluza^TM^ analysis for a hematopathologist’s final review. This process is time-consuming and takes at least 10–20 min per sample [11,12]. As manual analysis is susceptible to substantial inter-operator variation, the quality of the analysis results can differ. Given the current workforce challenges in clinical laboratories, this practice further burdens pathology departments and laboratories, hindering accurate diagnosis and negatively impacting clinical decisions [13].

Artificial intelligence (AI) is the term used to describe the intelligent behavior of a computer to achieve human-level performance in cognition-related tasks [14]. AI-based methods have demonstrated strong potential to enhance the sensitivity and specificity of assays in cancer detection, image pattern recognition, and a large repertoire of data processing tasks [15,16,17,18]. In hematopathology, AI has been used in blood cell classification, cytogenetic karyotyping, and therapeutic response evaluation [18]. Several publications have demonstrated the feasibility of AI-assisted flow cytometry in acute leukemia, classic Hodgkin lymphoma, and mature B-cell lymphoma diagnosis [12,19,20,21,22,23,24,25,26]. All these studies have consistently shown high accuracy in validation and testing, irrespective of the algorithms employed. However, none of the published articles are related to immunological disorders. Additionally, a significant knowledge gap exists in applying these models clinically, using routine daily samples beyond the initial validation step. Hence, there is an urgent need not only to develop effective AI models but also to demonstrate their practical application in clinical settings.

In this study, we developed and validated an AI-assisted flow cytometry workflow utilizing the DeepFlow^TM^ software (version 2.1.1), specifically designed for the ALPS panel. The workflow was tested on a dataset comprising 379 clinical cases collected in the year 2021. The DeepFlow^TM^ software seamlessly imports LMD files from the flow cytometer to the server, performs automated analysis, and generates a comprehensive flow cytometry analysis report. This automatically generated report encompasses crucial parameters such as cell viability; immunophenotypes; and the cell count and percentage of T, B, and NK cells, as well as normal and abnormal lymphoid subsets, providing a preliminary diagnosis. In addition to the fully automated workflow, the DeepFlow^TM^ software offers an interactive mode, enabling hematopathologists to analyze rare outlier cases that the AI model may not have encountered during its training phase. This interactive mode allows for the manual modification of gating when necessary. The implementation of this AI-assisted workflow significantly enhanced efficiency, reducing the analysis time for each case to less than 5 min. Moreover, it minimized the variation associated with manual data in flow cytometry.

The DeepFlow^TM^ software demonstrates accurate clustering and differentiation of cell lineages, including T, B, and NK cells, as well as important immune cell subsets such as CD8+ cytotoxic T cells, CD4+ helper T cells, CD3+ DNTs, and class-switched or non-switched B cells. Comparative analysis with manual (Kaluza^TM^) analysis, using pathologist-determined cell percentages as the gold standard, reveals a strong correlation (*r* > 0.9) across the range of cell subsets. Our institutional experience highlights the clinical validation of this automated flow cytometry analysis algorithm, showcasing its potential to not only increase laboratory workflow efficiency but also enhance the consistency of data interpretation. The introduction of a simple, efficient, and automatic AI-assisted system ensures standardized, high-quality flow results, leading to accurate diagnoses and increased productivity. Importantly, this automated algorithmic approach holds promise in addressing healthcare disparities, particularly in small community laboratories where skilled technologists may be limited.

## 2. Materials and Methods

### 2.1. Study Design

This study was approved by the institutional review board (IRB) at Oregon Health & Science University (OHSU, protocol #22165). All clinical cases, totaling 379, that underwent ALPS flow cytometry panels in the year 2021 (1 January 2021–31 December 2021) were included in the study. All clinical cases have been reviewed and signed out by hematopathologists within the OHSU Department of Pathology. The cases were divided into three sets—training, validation, and testing sets—for AI development and validation, performed in sequential order. The training sets comprised 60 cases from the months of March and November. The validation sets included 133 cases from the months of January, February, June, and July. The testing sets comprised 186 cases from the months of April, May, August, September, October, and December.

### 2.2. Flow Cytometry

The flow cytometry analysis panel follows the standard operating procedure of the OHSU Pathology Laboratory Services. A minimum of 1 mL of whole blood was collected in an EDTA tube and processed within 24 h. Red blood cells (RBCs) were lysed after incubation with a lysing solution (BD BioSciences, Franklin Lakes, NJ, USA) for 15 min at room temperature. After two rounds of centrifugation, removing supernatants, and resuspending the cell pellet with phosphate-buffered saline (PBS), the leukocytes were divided into three tubes and incubated with fluorochrome-conjugated antibodies for 15 min in the dark at room temperature, following the manufacturer’s instructions. Tube 1 (ALPS-T tube) included CD3, CD4, CD8, CD19, CD25, CD45, CD127, TCR αβ, and TCRγδ. Tube 2 (ACT-T tube) included CD3, CD4, CD8, CD45, CD69, CD45RO, CD45RA, CD279 (PD1), and HLA-DR. Tube 3 (ALPS-B tube) included CD3, CD5, CD19, CD27, CD45, CD56, IgD, kappa, and lambda. Flow cytometry data were acquired with a 10-color Navios^TM^ flow cytometer. A minimum of 100,000 events were collected. After data acquisition was completed, technologists transferred the generated LMD files to the designated driver for analysis using Kaluza^TM^ analysis software (version 2.2.1). All of the antibodies, equipment, and software are from Beckman Coulter (Brea, CA, USA).

### 2.3. AI-Assisted Flow Cytometry Analysis Workflow

DeepFlow^TM^ is an AI-assisted flow cytometry analysis software developed by DeepCyto LLC, based in West Linn, OR, USA. The DeepFlow^TM^ AI workflow comprises five sequential data processing stages, as illustrated in Figure 1. In the initial step, termed data validation, the software conducts an integrity check on the input LMD file, examining signal levels to eliminate abrupt fluctuations in flow, voltage, or experimental conditions. The identification and exclusion of doublets involve a linear regression model, delineating a linear separator for single cells versus doublets in forward scatter area (FSC-A) and forward scatter height (FSC-H). Subsequently, debris removal is executed based on low FSC-A/low sider scatter area (SSC-A), thereby concluding the initial cleanup of LMD data. The second step involves the utilization of a proprietary Automatic Gating Engine (AGE) to identify cell populations. The AGE conducts bottom-up clustering across all dimensions concurrently, considering cell distribution density and marker expression phenotype on all markers, categorized into five levels: bright, positive, partial, dim, and negative. Cell populations are defined when both criteria are satisfied. In the subsequent cell classification step, a proprietary Automatic Phenotype Classifier (APC), employing an AI model, categorizes identified cell populations into common groups (e.g., lymphocytes or B cells) and cell populations of interest. Specifically, the APC engine uses a multidimensional density–phenotype coupling (MDPC) algorithm for flow cytometry data, applied to all nucleated cells. The algorithm simultaneously considers all channels of flow cytometry, adjusting a cluster’s span based on the overall distribution. Two main criteria differentiate cell populations: distribution density across markers and marker expression (median fluorescent intensity and relative expression level). Pervasive expression levels are classified as bright, positive, partial, dim, and negative. New cell populations form when these criteria are met. At the end, all nucleated cells are grouped into clusters. At this stage, the APC takes over to build a random forest classifier with bootstrap aggregating to classify cell clusters into different categories. The incremental training of the cell differentiation AI model can utilize manually gated LMD files based on panels with a similar antibody setup on different reagents or color schemes. During the AI-assisted report phase, the information acquired earlier is utilized to calculate the percentage of each cell lineage. Ultimately, this information is summarized in a PDF report, including cell viability; immunophenotypes; the cell count and percentage of T, B, and NK cells; normal and abnormal lymphoid subsets; preliminary diagnoses; 2D scatterplots illustrating each intermediate analysis step; and quality control metrics.

### 2.4. Comparison of AI Results with Manual Results 

The percentage of cell populations in each case is determined by board-certified hematopathologists via manual analysis using Kaluza^TM^ software (version 2.2.1) and is considered the gold standard. The Pearson correlation coefficient between manual and AI results by DeepFlow^TM^ was compared using linear regression analysis with SPSS Statistics, Version 28.0 (IBM Corp, Armonk, NY, USA). The comparison data were demonstrated as 2D correlation plots, and an *r*-value > 0.90 was considered a strong correlation. 

## 3. Results

### 3.1. AI Software Enhances Automation of Flow Cytometry Analysis 

The first step in AI validation is to verify the AI software (DeepFlow™, version 2.1.1) workflow applicable to our system. The AI workflow commences when the LMD acquisition files are completed by the flow cytometer and concludes with hematopoietic cell gating and analysis (Figure 1). The entire process is fully automated, as manual steps, such as case folder creation and file transfer to designated drives, are all replaced by AI software. The AI software also performs automatic gating and generates preliminary diagnostic reports for hematopathologists to review. Importantly, the software includes a feature that facilitates manual fine-tuning for gating. This human intervention can contribute to the continuation of incremental AI model training. Compared to the conventional workflow, AI was able to complete the data analysis portion of the process in less than 5 min, a task that typically takes a technologist 10–20 min.

The strength of AI software lies in its resilient self-learning capability. Preceding this version, the existing model underwent iterative training encompassing over 50,000 flow cytometry cases. In this investigation, an algorithm specific to the ALPS panel was established using a training set of 60 cases. Following training, a validation set of 133 cases underwent testing for fine-tuning adjustments. Subsequently, the testing set, comprising the largest number of cases (186 cases), was processed using the most recent AI software version. The detailed algorithm workflow is illustrated in Figure 2.

### 3.2. AI Classifies Cells Accurately Comparing to Kaluza^TM^ Analysis 

Proper cell classification is fundamental for achieving accurate flow cytometry analysis. Employing manual analysis with Kaluza^TM^ software as the gold standard, the DeepFlow^TM^ AI accurately classified T, B, and NK cells, as well as vital immune cell subsets, including CD8+ cytotoxic T cells, CD4+ helper T cells, CD45RA+ T cells, CD45RO+ T cells, kappa and lambda light chains, and class-switched or non-switched B cells (Figure 3). When it comes to disease conditions, AI can also recognize and classify normal and abnormal subsets. Figure 4 illustrates a case of an ALPS patient presenting with lymphocytosis and lymphadenopathy. The DNT population, comprising approximately 30% of the total CD3+ T cells, was identified in the left lower quadrant. This percentage exceeds the essential criterion of 2.5% of CD3+ lymphocytes. In a separate case, AI identified a significantly decreased CD45RA+ level, approximately 8% (reference range 30–60%), in the first plot, a corresponding increase in CD45RO+ level in the second plot, and a decreased count of switched memory B cells at <2% (reference range >5%). The overall pattern is compatible with DiGeorge syndrome (Figure 5).

### 3.3. AI Demonstrated Strong Correlation for Cell Enumeration Compared to Kaluza^TM^ Analysis 

The enumeration and percentage of cells are crucial for diagnosing diseases, as the threshold of cells may determine the diagnosis of different entities. The Pearson correlation coefficient was calculated for each cell subtype between manual Kaluza^TM^ and DeepFlow^TM^ AI results. The correlation plots demonstrated strong correlations (*r* > 0.9 or very close to 0.9) in both T, B, and NK cells, as well as their subsets, across all three study sets (Figure 6 and Figure 7). Such a strong correlation verifies the accuracy of AI-assisted flow analysis. 

## 4. Discussion 

In this study, we developed and validated an automatic AI-assisted flow cytometry software (DeepFlow™) using 379 clinical cases with a flow panel designed for ALPS and other immunological disorders. The cases were derived from real-world clinical patients whose blood samples were collected and tested with the ALPS panel in 2021. The AI software accurately classified and enumerated T, B, and NK cells, along with important immune cell subsets, demonstrating a strong correlation (*r* > 0.9) compared to manual Kaluza™ analysis. The AI software is fully automated and significantly improves the workflow, reducing turnaround time by 10 times and minimizing the subjectivity of human gating. The manual interactive feature in the software provides the flexibility of human intervention, which is crucial as AI may encounter rare cases it never encountered during AI training. This study highlights the accuracy and efficiency of an AI-assisted flow cytometry diagnosis for immunological disorders in a clinical setting.

Since the concept of AI was first described in 1956, its impact on medicine has been expanding [27]. The application of AI in healthcare can be broadly grouped into four categories: imaging analysis, improved workflow and efficacy, public health-related fields, and big data processing [28]. Currently, the US Food and Drug Administration (FDA) has approved at least 29 AI-based medical devices and algorithms across different medical specialties. These include radiograph interpretation, glucose level management in diabetic patients, electrocardiogram analysis, sleep disorder diagnosis, and others [29]. In the field of surgical pathology, the advancement of whole-slide imaging, where entire glass slides are scanned and digitized at high resolution, creates great potential for clinical implications. A landmark AI-based software, Paige Prostate, was the first software approved by the FDA in 2021 to assist pathologists in the detection of cancer in prostate biopsies [30]. Multiple studies from different institutions confirmed the diagnostic accuracy of using AI for prostate cancer detection and improvement in pathologists’ efficiency [31,32]. 

Conventional flow cytometers have been the mainstay in hematopathology for decades. However, emerging technologies like spectral flow cytometry and mass flow cytometry have gained significant attention in research settings. In recent years, these technologies have started to be utilized in clinical laboratories in some countries [33]. In a conventional flow cytometer, individual lasers are equipped with a series of light filters positioned in front of photon detectors. In contrast, spectral flow cytometry uses many detectors to collect all the emitted light from each fluorophore by each laser [33]. Therefore, spectral flow cytometry can detect more fluorophores simultaneously, offering greater flexibility in assay design. Mass cytometry even allows for the detection of up to 50 parameters in one sample because antibodies are labeled with metal isotopes and separated based on their mass, and there is no fluoresce spillover like conventional flow cytometry [34]. While spectral flow cytometry has been mainly used in research settings, early adaption studies have shown very promising applications in clinical settings to detect immune cells in patients with myelodysplastic syndrome (MDS) [35]. 

The new emerging flow cytometry technology demands an enhanced ability to handle high-dimensional data. When confronted with modern high-dimensional flow cytometry data, especially for spectral flow cytometry and mass cytometry, the dimension of flow cytometry data has transitioned from the conventional 8–10 colors per tube to exceeding 50 colors. Traditional two-dimensional plots often prove inadequate in representing the intricate high-dimensional structures inherent in flow cytometry data. In addition to the subjective biases introduced by manual gating analysis, consistently measuring the numerical expression levels of various antigens in the multidimensional space of the target cell group proves to be challenging [36]. In addition to the limited capacity of data processing, manual gating also presents other drawbacks, including low processing speed and susceptibility to human errors. It is typical for various human experts to obtain differing gating results for identical flow cytometry data. This human-induced variability presents a significant obstacle to maintaining the quality control of clinical data. 

Therefore, it is crucial to develop a flow cytometry analysis methodology that minimizes the effect of human elements. This would allow for the concurrent evaluation of numerous cell populations and their antigen expression levels in a multidimensional context, resulting in a more reliable and objective interpretation of flow cytometry data. To address the challenges associated with manual gating, numerous computational tools have been developed to automate each step of flow cytometry data analysis. These steps encompass data quality control, normalization, visualization, cell population identification, and sample classification. These tools employ various algorithms, ranging from rule-based algorithms to artificial intelligence (AI) models. The rapid advancement of AI technology and its applications in medicine in recent years has rendered this goal attainable. Various general-purpose algorithms for dimension reduction and clustering, such as T-SNE and K-means, have been utilized to address the analysis of MFC data [37,38]. Furthermore, several MFC-specific algorithms, including SPADE, FlowSOM, and PhenoGraph, have been devised for processing MFC data [39,40]. Besides the AI algorithm in MFC, the use of AI models in the diagnosis of hematological malignancies is also rapidly evolving [41,42].

Currently, automation algorithms are mainly designed for research purposes, limiting their use in clinical settings due to slow performance and challenges in result interpretation. Comparing automated outcomes to manual analyses is difficult, making their application more challenging. For example, the widely used T-SNE-based unsupervised clustering algorithms in flow cytometry publications are time-consuming, and it takes hours to analyze common MRD panels with a million events, making them impractical for clinical use. Additionally, the stochastic nature of the T-SNE approach introduces variability in clustering results with each run, complicating the understanding of cell populations for clinicians using the same flow data source. This inconsistency poses a significant obstacle to integrating such algorithms into clinical practice.

Several AI algorithms and machine learning applications for clinical flow cytometry have been developed and reported, covering a broad spectrum, from mature B-cell neoplasm subtyping and acute leukemia classification to detecting residual diseases in acute myeloid leukemia and MDS [20,24,43,44,45]. These heuristic algorithms are meticulously crafted to suit specific flow panels, making them challenging to generalize for handling more generic flow panels.

One challenge that AI models currently face is that MFC data are often limited by the specific panel on which they underwent training. There is no standardization regarding which antibodies should be used and how each plot will be displayed, as MFC panels are generally laboratory-specific. Therefore, a significant challenge in integrating AI into routine diagnostics lies in the robustness and adaptability of these models. The MDPC algorithm employed in the DeepFlow^TM^ software offers a robust solution for generic flow cytometry panels in clinical settings. Its transfer learning AI model minimizes the need for an extensive training set, as MDPC dynamically adapts to variations in reagents and instrument settings. This adaptability allows MDPC to accurately capture the relative expression of CD markers and differentiate various cell lineages efficiently.

As evidenced in this study, the DeepFlow^TM^ workflow is not intended to replace human analysis. Instead, its design aims to offer an objective and consistently reliable second opinion analysis, serving as a valuable reference for clinicians. Notably, the DeepFlow^TM^ software has been successfully used to diagnose acute leukemia in a different academic medical center [12]. In this study, DeepFlow^TM^ clearly demonstrated its robust machine learning ability and easily transferable features among different flow panels in various institutions. The incremental training model offers clinical institutions the possibility to start with relatively small-scale training datasets while maintaining strong performance. This advantage of easy adaptability will be specific for non-academic laboratories considering the adoption of AI, especially when experienced staff and labor are limited. 

The efficiency of AI compared to human labor is evident; our calculations suggest that a lab technologist would require roughly 20 min to finalize a case. Nevertheless, this approximation might be understated in an actual clinical environment, as technologists frequently upload and examine cases in batches to boost overall productivity. In these instances, the initial cases could have been finished about an hour earlier. The standardization and automation of pipeline workflow in data acquisition, analysis, and reporting are the key elements in increasing the accuracy and effectiveness of diagnostic testing, ultimately striving to provide the best patient care [46]. 

One obstacle to using AI in clinical settings is that the algorithm can be hard to understand, and the information can be intimidating for lab personnel without a programming background. Our approach to addressing this challenge is implementing Kaluza^TM^ software as the standard and back-gating similar or identical diagnostic plots, illustrating to the hematopathologist how the model operates. This method, involving plot-to-plot comparisons and cell population correlations, offers a direct visualization presentation of AI analysis results for hematopathologists, thereby alleviating the stress associated with the steep learning curve of any new software.

A notable challenge we encountered in this study involved the correlation of a small percentage of cells, such as myeloid dendritic cells and plasmacytoid dendritic cells. Classifying these cells requires calculations from different subsets across various plots and is particularly prone to significant variation, where even a small gate change can cause a prominent downstream effect. Currently, we are fine-tuning the algorithm to enhance the consistency of these sparse cell populations. Another limitation of using this algorithm is its inability to recognize patterns it has not been trained on. Continuous training is necessary to update the AI, enabling it to identify new patterns as new markers develop and new immunophenotypes are identified.

In summary, we have successfully validated the DeepFlow^TM^ software with real-world clinical specimens for the ALPS panel. The AI software has demonstrated its fast and consistent performance for MFC data analysis and interpretation. Our ongoing efforts involve expanding this study to validate other flow panels in our practice and establishing a comprehensive clinical decision-making assistant system.

## Figures and Tables

**Figure 1 diagnostics-14-00420-f001:**
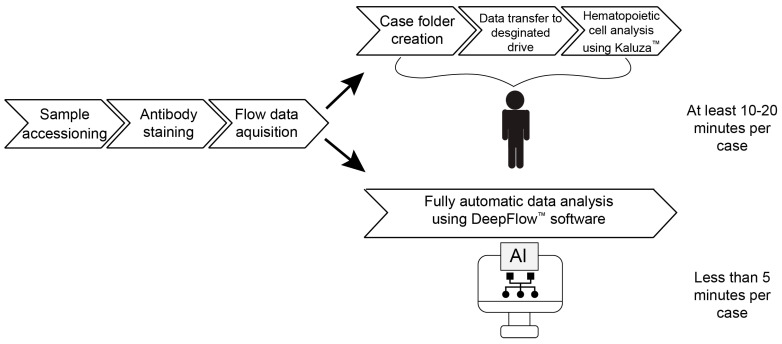
The comparison of conventional and AI-assisted flow cytometry workflow. The conventional workflow involves technologists manually uploading and analyzing MFC plots. The AI software (version 2.1.1) is fully automated and can save the processing time to less than 5 min per case.

**Figure 2 diagnostics-14-00420-f002:**
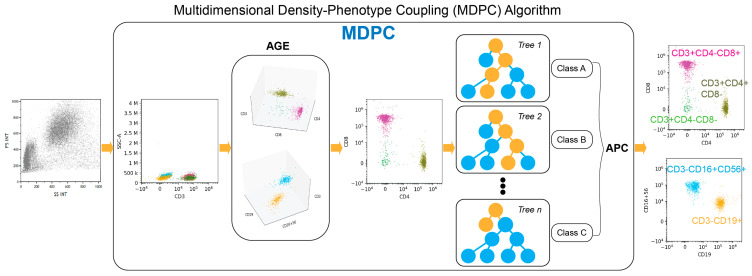
The MDPC AI algorithm was utilized in this study, comprising two main components (AGE and APC). The first step involves using unsupervised learning to group cells from MFC data into clusters (AGE step). This model employs incremental training, utilizing a large dataset trained before this study and enabling the AI to recognize cells based on their immunophenotypes. The second part of the algorithm distinguishes cell lineages among the clustered cells based on their multidimensional immunophenotypes, such as the presence or absence of cell markers and strong or weak expression of certain markers, ultimately generating a new plot (APC step). The AI-generated plots will be reviewed by pathologists for manual gating and adjustments, if needed, before final reporting.

**Figure 3 diagnostics-14-00420-f003:**
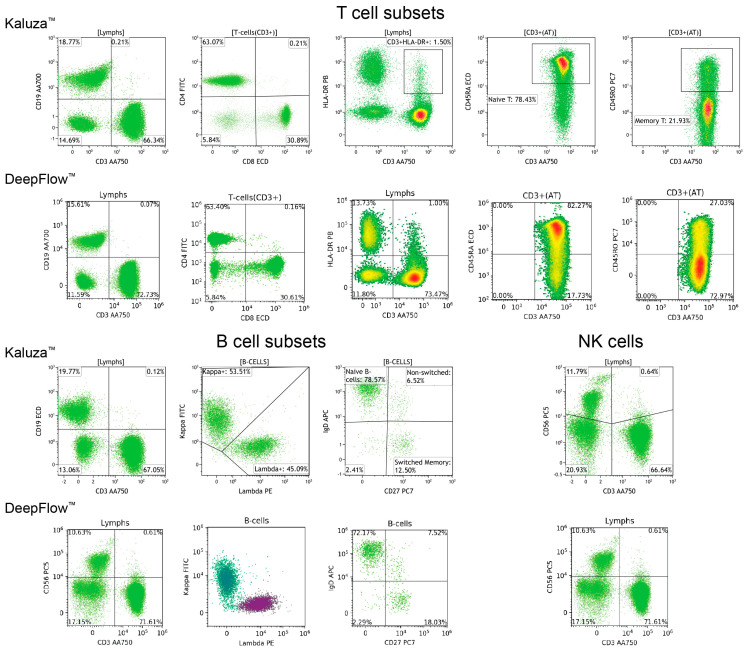
Comparison of MFC plots generated by Kaluza^TM^ and AI DeepFlow^TM^ for normal T, B, and NK cells.

**Figure 4 diagnostics-14-00420-f004:**
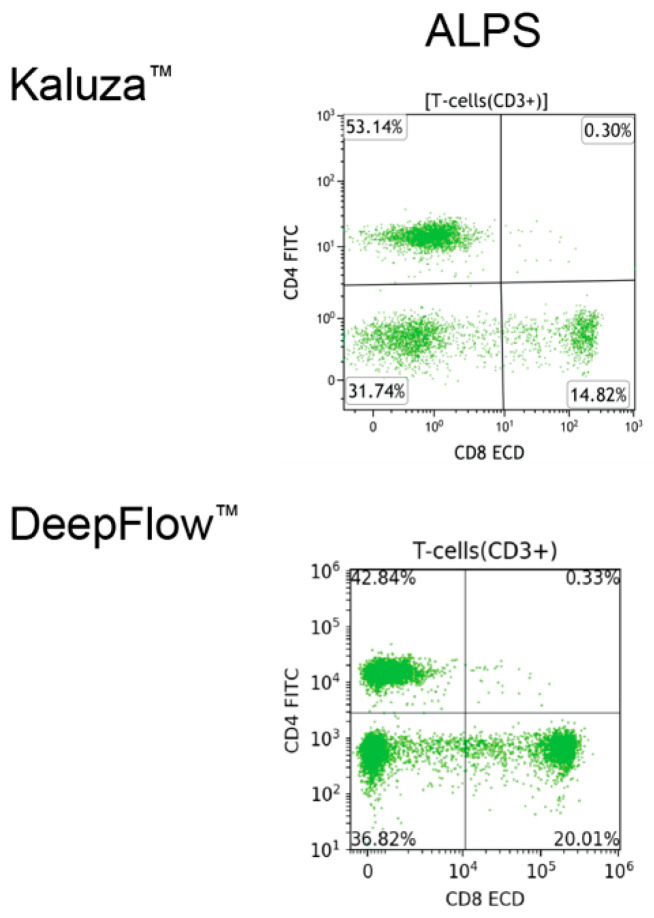
A comparison of MFC plots generated by Kaluza^TM^ and AI DeepFlow^TM^ for a patient with ALPS reveals that DNTs in the left lower quadrant constitute approximately 30% of the total CD3+ T cells.

**Figure 5 diagnostics-14-00420-f005:**
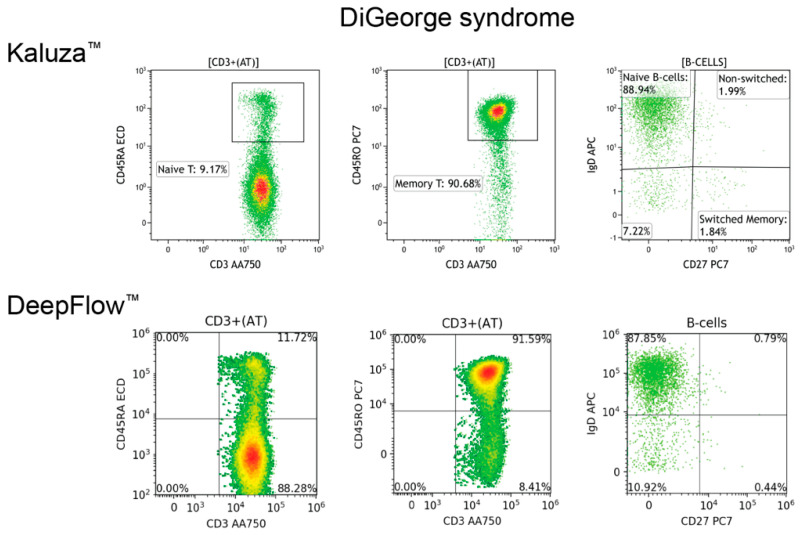
A comparison of MFC plots generated by Kaluza^TM^ and AI DeepFlow^TM^ for a patient with DiGeorge syndrome reveals a significantly decreased CD45RA+ population highlighted in the box of the first Kaluza^TM^ plot. Correspondingly, an increased CD45RO+ population is observed in the box of the second plot. Additionally, the right bottom quadrant of the third plot shows a decreased count of switched memory B cells.

**Figure 6 diagnostics-14-00420-f006:**
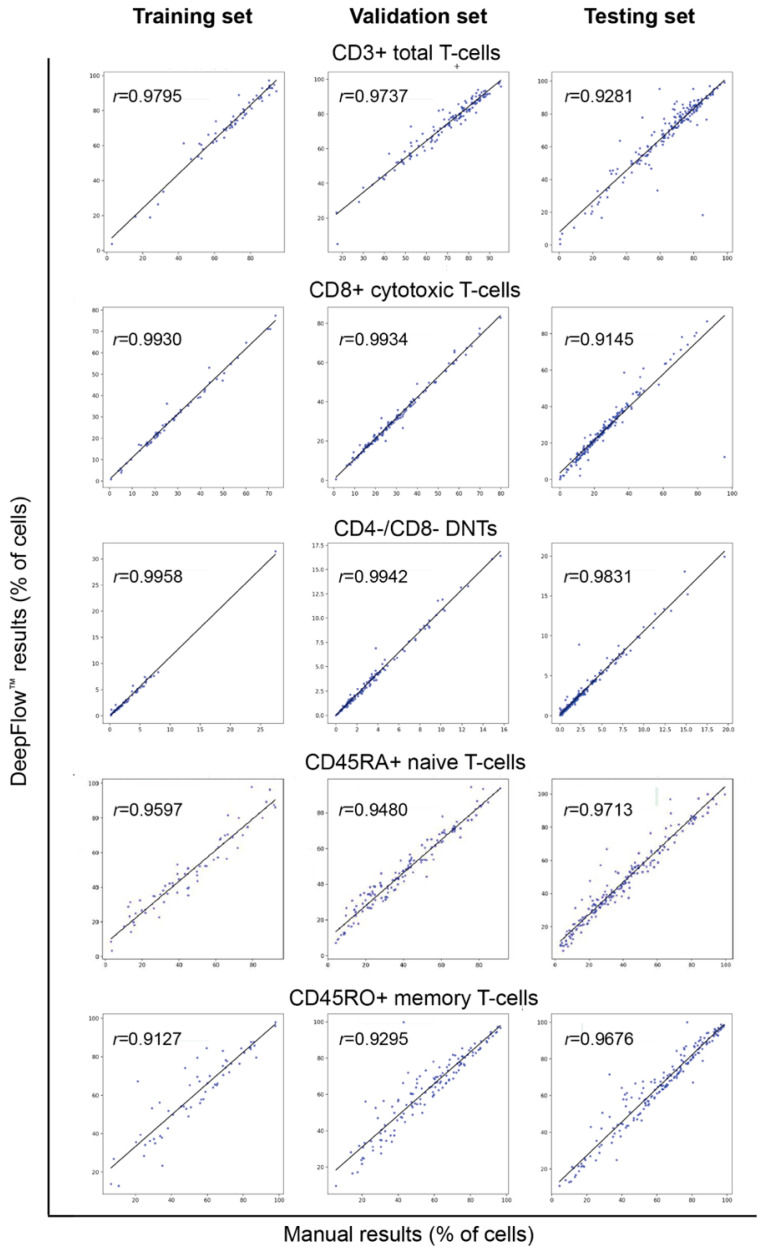
Comparison of correlation coefficients of essential T-cell subsets between manual analysis using Kaluza^TM^ and AI analysis by DeepFlow^TM^ reveals correlation coefficients greater than 0.9 for all three sets.Each dot represents one case.

**Figure 7 diagnostics-14-00420-f007:**
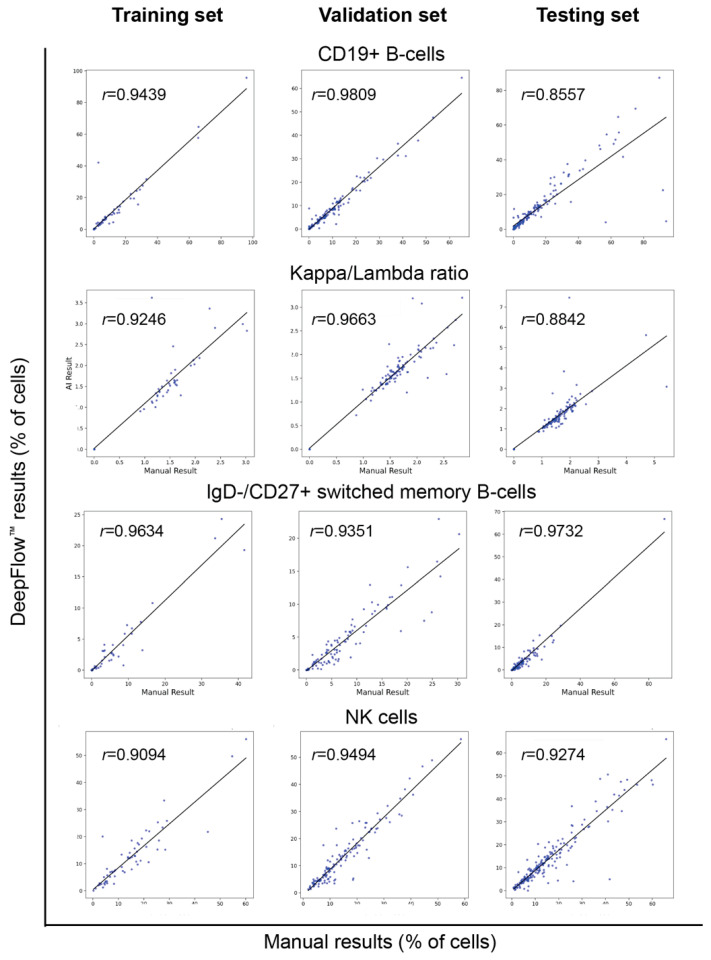
Comparison of correlation coefficients of essential B- and NK-cell subsets between manual analysis using Kaluza™ and AI analysis by DeepFlow™ reveals that the majority of correlation coefficients in both B and NK cells are either greater than or close to 0.9. Each dot represents one case.

## Data Availability

The data and material in our studies are available upon request to the corresponding author.

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
