# Peer review of "Validation of Artificial Intelligence (AI)-Assisted Flow Cytometry Analysis for Immunological Disorders"

_diagnostics, 2024, doi:10.3390/diagnostics14040420_

Round 1

Reviewer 1 Report

Comments and Suggestions for Authors

The article entitled “Validation of artificial intelligence (AI)-assisted flow cytometry analysis for immunological disorders” is well-written and, from my point of view, would be of interest for the readers of Diagnostics. In spite of this, and before its publication, I consider that authors should perform the following changes:

Introduction: at the end of the introduction a description of the layout of the manuscript indicating the sections and their contents would be helpful for readers.

Figure 1: I recommend splitting this figure in two parts and giving a more detailed explanation in the text of the manuscript.

Table 1 can be removed as the information that it contains is also in the text.

Author Response

Reviewer 1:

The article entitled “Validation of artificial intelligence (AI)-assisted flow cytometry analysis for immunological disorders” is well-written and, from my point of view, would be of interest for the readers of Diagnostics. In spite of this, and before its publication, I consider that authors should perform the following changes:

Introduction: at the end of the introduction a description of the layout of the manuscript indicating the sections and their contents would be helpful for readers.

Authors response: We appreciate the reviewer for the positive comments.

  1. Figure 1: I recommend splitting this figure in two parts and giving a more detailed explanation in the text of the manuscript.

Authors response: We appreciate this suggestion and have made changes accordingly and split the Fig 1 into two different figures.

Action: The figures initially labeled as Fig 1A and 1B have been changed to Fig 1 and Fig 2. Additionally, a lay-language description has been added to the figure legend of Fig 2 to facilitate better understanding.

Lines: 196-204: The MDPCI AI algorithm was utilized in this study, comprising two main components (AGE and APC). The first step involves using unsupervised learning to group cells from MFC data into clusters (AGE step). This model employs incremental training, utilizing a large dataset trained before this application, enabling the AI to recognize cells based on their immunophenotypes. The second part of the algorithm distinguishes cell lineages among the clustered cells based on their multidimensional immunophenotypes, such as the presence or absence of cell markers, strong or weak expression of certain markers, ultimately generating a new plot (APC step). The AI-generated plots will be reviewed by pathologists for manual gating and adjustments, if needed, before final reporting.

  1. Table 1 can be removed as the information that it contains is also in the text.

Action: We appreciate this feedback and Table 1 is removed.

Reviewer 2 Report

Comments and Suggestions for Authors

The article focuses on the problem related to manual analysis of flow cytometry data is time-consuming and prone to variation, hindering its effectiveness in diagnosing immunological disorder. Study developed and validated an AI-assisted flow cytometry workflow to address the challenges.

This work demonstrates the potential of AI in significantly improving the efficiency and accuracy of flow cytometry for diagnosing immunological disorders. But there are few questions which need to be addressed, it is  missing in the article.

·       What are the research gaps from previous work? It’s not covered, authors must add in literature section to brief on the strength and weakness of previous study and how the gaps are addressed in the current study?

·       What are the limitations of the current AI model?

·       How robust is the model to variations in data processing and acquisition?

·       What are the ethical considerations of using AI in clinical diagnostics?

·       Are there plans for further studies to evaluate the long-term impact of AI-assisted flow cytometry in clinical practice?

While correlation with manual analysis is important, but additional studies could explore the clinical impact of AI-assisted diagnoses, such as agreement with clinical outcomes.

Authors can explain the understanding how the AI model reaches its conclusions and its limitations in specific cases can build trust and transparency for clinicians.

Results are promising, the use of a large clinical dataset and comparison to manual analysis with high correlation (r > 0.9) demonstrate the AI's effectiveness.

Overall, this study presents a compelling case for AI in flow cytometry analysis and highlights its potential to revolutionize the field. Addressing the questions mentioned above could further strengthen the study and quality of the article

Comments on the Quality of English Language

Typos and grammer can be thoroughly checked

Author Response

Reviewer 2:

Authors response: We appreciate the reviewer for the positive comments and constructive suggestions.

The article focuses on the problem related to manual analysis of flow cytometry data is time-consuming and prone to variation, hindering its effectiveness in diagnosing immunological disorder. Study developed and validated an AI-assisted flow cytometry workflow to address the challenges.

This work demonstrates the potential of AI in significantly improving the efficiency and accuracy of flow cytometry for diagnosing immunological disorders. But there are few questions which need to be addressed, it is missing in the article.

  1. What are the research gaps from previous work? It’s not covered, authors must add in literature section to brief on the strength and weakness of previous study and how the gaps are addressed in the current study?

Authors response: Thanks for bringing up this great point. Firstly, there are only a handful of papers in clinical flow cytometry focusing on the development of AI models for the diagnosis of different hematological diseases. Generally, these papers demonstrated strong accuracy in validation and testing, regardless of the algorithm used and the targeted diagnostic categories. However, as the field is still in its infancy, no winning algorithm can be declared in any of the studies, including ours. Secondly, none of the papers are validating immunological disorders, and ours is the first one to do so. Thirdly, one of the most significant gaps missing from previous studies was the lack of expansion beyond the initial model validation. In our study, we focused on clinical validation using our daily practice samples. This is the first time we have concentrated on clinical application instead of just model validation.

Action: We have included additional references in the introduction related to the current AI and flow cytometry studies (references #21-#26) in line 85.1-6 Additionally, we have added the following statements in the introduction section (lines 85-91) to clarify the goal of this study.

Lines 85-91: All these studies have consistently shown high accuracy in validation and testing, irrespective of the algorithms employed. However, none of the published articles are related to immunological disorders. Additionally, a significant knowledge gap exists in applying these models clinically, using routine daily samples beyond the initial validation step. Hence, there is an urgent need not only to develop effective AI models but also to demonstrate their practical application in clinical settings.

  1. What are the limitations of the current AI model?

Authors response:  The main limitation of the current AI model is its inability to recognize patterns it hasn't been trained on. In one of our ongoing projects evaluating acute leukemia diagnosis for acute lymphoblastic leukemia (ALL) and acute myeloid leukemia (AML), we found that AI can recognize and diagnose ALL and AML with CD34+ blast population, showing good concordance with hematopathologists. However, AI failed to recognize blast populations that are CD34 negative.  This highlights that the AI algorithm is a dynamic and evolving program. As new markers develop and new immunophenotypes are identified, continuous training is necessary to update the AI, enabling it to recognize new patterns.

Action: We have added the following statements to the discussion section. These added paragraphs address current limitation in other AI algorithms and highlight limitations specific to our MDPC algorithm.

Lines 363-372: Currently, automation algorithms are mainly designed for research purposes, limiting their use in clinical settings due to slow performance and challenges in result interpretation. Comparing automated outcomes to manual analyses is difficult, making their application more challenging. For example, widely-used T-SNE-based unsupervised clustering algorithms in flow cytometry publications are time-consuming, taking hours to analyze common MRD panels with a million events, making them impractical for clinical use. Additionally, the stochastic nature of the T-SNE approach introduces variability in clustering results with each run, complicating the understanding of cell populations for clinicians using the same flow data source. This inconsistency poses a significant obstacle to integrating such algorithms into clinical practice.

Lines 420-423: Another limitation of using this algorithm is its inability to recognize patterns it hasn't been trained on. Continuous training is necessary to update the AI, enabling it to identify new patterns as new markers develop and new immunophenotypes are identified.

Additionally, we have added a few paragraphs in lines 376-378 and 389-391 to further clarify contents in the discussion.

Lines 376-378: These heuristic algorithms are meticulously crafted to suit specific flow panels, making them challenging to generalize for handling more generic flow panels.

Lines 389-391: As evidenced in this study, the Deepflow™ workflow is not intended to replace human analysis. Instead, its design aims to offer an objective and consistently reliable second opinion analysis, serving as a valuable reference for clinicians.

  1. How robust is the model to variations in data processing and acquisition?

Authors response:  Our algorithm process data by dynamic range to determine the immunophenotype. Therefore, this dynamic range offers greater tolerance and flexibility regardless of reagent or instrumental setting changes.

Action: We have added a paragraph to further explain the robustness.

Lines 383-388: The MDPC algorithm employed in the Deepflow™ software offers a robust solution for generic flow cytometry panels in clinical settings. Its transfer learning AI model minimizes the need for an extensive training set, as MDPC dynamically adapts to variations in reagents and instrument settings. This adaptability allows MDPC to accurately capture the relative expression of CD markers and differentiate various cell lineages efficiently.

  1. What are the ethical considerations of using AI in clinical diagnostics?

Author response: While AI presents new opportunities for improving patient care, the ethical aspect of AI is an area that has not been thoroughly investigated yet. Regardless, like any other medical equipment or tool to deal with patient samples, there are certain rules to keep in mind.

  • Accuracy and Reliability: AI systems must be highly accurate and reliable in diagnosing medical conditions. This is particularly important for clinical laboratory medicine, as our goal is to make an accurate diagnosis so that patients can receive proper treatment.
  • Transparency and Explainability: AI algorithms should be transparent and explainable so that healthcare providers and patients can understand how a diagnosis or recommendation was made. Clinicians need to understand how the AI works and be able to explain it to patients. Black-box AI systems can raise concerns about accountability and trust.
  • Bias and Fairness: AI models can inherit biases present in the data they are trained on. Biased algorithms can lead to disparities in healthcare outcomes, particularly affecting marginalized or underrepresented groups. In our study, we used whole-year clinical cases to conduct the validation to minimize selection bias.
  • Human Oversight: While AI can assist healthcare professionals, it should not replace them entirely. In our study, hematopathologists have final control over the gating and reporting.
  • Access and Equity: The availability and affordability of AI-assisted diagnostic tools should be equitable to avoid creating disparities in healthcare access. One of the biggest advantages of AI is that it provides equal healthcare to rural community hospitals or laboratories where they might be short of experienced personnel or just short-staffed.
  • Data Privacy and Security: Any patient data strictly follows HIPPA rules. In our study, we followed our institutional security measures to handle patients' data. All cases are de-identified for this study after our robust institutional security check.

In summary, we took into account a lot of considerations when designing AI research. We believe a highly regulated AI tool can help improve the quality of patient care.

  1. Are there plans for further studies to evaluate the long-term impact of AI-assisted flow cytometry in clinical practice?

Authors response:  Yes, absolutely! We are working on multiple projects related to AI-assisted flow cytometry using different panels, including leukemia panel, lymphoma panel and multiple myeloma panels. We are working on develop a pipeline to incorporate the AI to assist majority if not all of our panels.

  1. While correlation with manual analysis is important, but additional studies could explore the clinical impact of AI-assisted diagnoses, such as agreement with clinical outcomes.

Authors response:   In this study, we have demonstrated that AI can assist in the diagnosis of immunodeficiency and autoimmune disorders, such as DiGeorge syndrome and ALPS. Furthermore, in one of our ongoing projects evaluating acute leukemia diagnosis for acute lymphoblastic leukemia (ALL) and acute myeloid leukemia (AML), we found that AI can recognize and diagnose ALL and AML with CD34+ blast population, showing good concordance with hematopathologists. Currently, we are working on training the AI model to recognize CD34- blast population.

  1. Authors can explain the understanding how the AI model reaches its conclusions and its limitations in specific cases can build trust and transparency for clinicians.

Authors response: Thank you for this great question. Regarding how AI reaches its conclusions, there are basically two main steps: The first step is to use unsupervised learning to group cells from MFC data into clusters (AGE step in Fig 2). Since this model utilizes incremental training, a large dataset was trained before this application, allowing the AI to learn how to recognize cells based on their immunophenotype. The second part of the algorithm differentiates cell lineage among the clustered cells based on their multi-dimensional immunophenotypes (strong or weak expression, etc.) and eventually generates a new plot (APC step in Fig 2). The AI-generated plots will be reviewed by pathologists for manual gating and adjustments, if needed, before final reporting.

Action: We have updated figure legend of Fig 2 to facilitate better understanding.

Lines: 196-204: The MDPC AI algorithm was utilized in this study, comprising two main components (AGE and APC). The first step involves using unsupervised learning to group cells from MFC data into clusters (AGE step). This model employs incremental training, utilizing a large dataset trained before this application, enabling the AI to recognize cells based on their immunophenotypes. The second part of the algorithm distinguishes cell lineages among the clustered cells based on their multidimensional immunophenotypes, such as the presence or absence of cell markers, strong or weak expression of certain markers, ultimately generating a new plot (APC step). The AI-generated plots will be reviewed by pathologists for manual gating and adjustments, if needed, before final reporting.

Results are promising, the use of a large clinical dataset and comparison to manual analysis with high correlation (r > 0.9) demonstrate the AI's effectiveness.

Overall, this study presents a compelling case for AI in flow cytometry analysis and highlights its potential to revolutionize the field. Addressing the questions mentioned above could further strengthen the study and quality of the article.

References:

  1. Seifert RP, Gorlin DA, Borkowski AA. Artificial Intelligence for Clinical Flow Cytometry. Clin Lab Med. Sep 2023;43(3):485-505. doi:10.1016/j.cll.2023.04.009
  2. Clichet V, Harrivel V, Delette C, et al. Accurate classification of plasma cell dyscrasias is achieved by combining artificial intelligence and flow cytometry. Br J Haematol. Mar 2022;196(5):1175-1183. doi:10.1111/bjh.17933
  3. Gaidano V, Tenace V, Santoro N, et al. A Clinically Applicable Approach to the Classification of B-Cell Non-Hodgkin Lymphomas with Flow Cytometry and Machine Learning. Cancers (Basel). Jun 24 2020;12(6)doi:10.3390/cancers12061684
  4. Ko BS, Wang YF, Li JL, et al. Clinically validated machine learning algorithm for detecting residual diseases with multicolor flow cytometry analysis in acute myeloid leukemia and myelodysplastic syndrome. EBioMedicine. Nov 2018;37:91-100. doi:10.1016/j.ebiom.2018.10.042
  5. Ng DP, Wu D, Wood BL, Fromm JR. Computer-aided detection of rare tumor populations in flow cytometry: an example with classic Hodgkin lymphoma. Am J Clin Pathol. Sep 2015;144(3):517-24. doi:10.1309/AJCPY8E2LYHCGUFP
  6. Simonson PD, Lee AY, Wu D. Potential for Process Improvement of Clinical Flow Cytometry by Incorporating Real-Time Automated Screening of Data to Expedite Addition of Antibody Panels. Am J Clin Pathol. Mar 3 2022;157(3):443-450. doi:10.1093/ajcp/aqab166
